# When Would Vision-Proprioception Policy Fail in Robotic Manipulation?

## Abstract

Proprioceptive information is critical for precise servo control by providing real-time robotic states. Its collaboration with vision is highly expected to enhance performances of the manipulation policy in complex tasks. However, recent studies have reported *confused observations* that vision-proprioception policies frequently suffer from poor generalization. In this work, we attempt to answer the question: **When would vision-proprioception policy fail?** To this end, we conducted temporally controlled experiments and found that during task sub-phases that robot's motion transitions, which require target localization, the vision modality of the vision-proprioception policy fails to take effect. Further analysis reveals that the policy naturally gravitates toward concise proprioceptive signals that offer faster loss reduction when training, thereby dominating the optimization and suppressing the learning of the visual modality during motion-transition phases. To alleviate this, we propose the Gradient Adjustment with Phase-guidance (GAP) algorithm that adaptively modulates the optimization of proprioception, enabling dynamic collaboration within vision-proprioception policy. Specifically, we leverage proprioception to capture robotic states and estimate the probability of each timestep in the trajectory belonging to motion-transition phases. During policy learning, we apply fine-grained adjustment that reduces the magnitude of proprioception's gradient based on estimated probabilities, leading to improved generalization of vision-proprioception policies. The comprehensive experiments demonstrate GAP is applicable in both simulated and real-world environments, across one-arm and dual-arm setups, and compatible with both conventional and Vision-Language-Action models. We believe this work can offer valuable insights into the development of vision-proprioception policies for robotic manipulation.

## 1 Introduction

Proprioceptive information has long been recognized as a cornerstone of low-level robotic control, enabling smooth motor behavior through immediate access to the robot's internal state. This capability is especially critical in tasks requiring high accuracy and fast correction, such as posture control Allum et al. (1998); Henze et al. (2014) and locomotion Bjelonic et al. (2016); Lee et al. (2020); Yang et al. (2023). In recent years, there has been growing interest in introducing proprioception to learning-based manipulation Levine et al. (2016); Cong et al. (2022); Jiang et al. (2025). Despite the expectations that its inclusion will empower manipulation policies to maintain precision and robustness across various scenarios, existing works have reported *confused observations*: HPT Wang et al. (2024) demonstrated clear improvements under the joint utilization of vision and proprioception, while Octo Octo Model Team et al. (2024) observed policies trained with additional proprioception seemed generally worse than vision-only policies. This discrepancy exposes a critical obstacle to understanding: **when vision-proprioception policy would fail in robotic manipulation?**

Submitted to 39th Conference on Neural Information Processing Systems (NeurIPS 2025). Do not distribute.

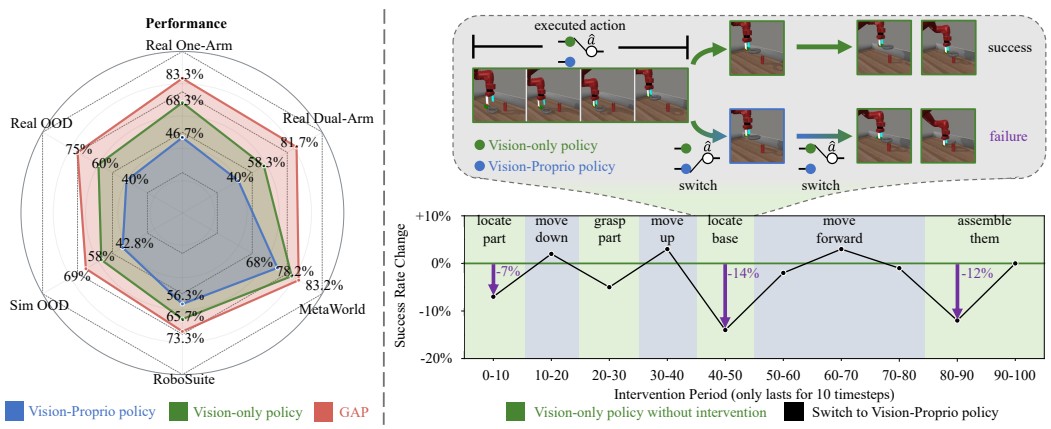

Figure 1: When would vision-proprioception policy fail? (left) Vision-Proprioception policies perform 15.8% worse than Vision-only policies. (right) We explore this through intervening the task execution of vision-only policy during different periods, by switching to vision-proprioception policy. Such intervention has minimal impact during motion-consistent phases like "move forward". However, during motion-transition phases like "locate base" and "assemble them", switching leads to noticeable degradation, indicating the vision modality fails to take effect during these phases.

Extensive prior studies have revealed that the importance of visual and proprioceptive information could change over time within manipulation Sarlegna and Sainburg (2009); Feng et al. (2024); He et al. (2025), which referred to as *Modality Temporality*. For example, during motion-consistent phases where the robot performs ongoing movements, the policy can benefit more from proprioceptive signals. In contrast, during the transition intervals where the robot's motion shifts, it is required to rely more on visual cues for accurate target localization. To verify whether the vision-proprioception policy exhibits such collaboration, we conduct an intervention experiment in the controlled simulation environment. Concretely, we execute the "assembly" task using the vision-only policy, but for a specific 10-timestep period, we replace executed actions with those predicted by the vision-proprioception policy under the same observations. As shown in Figure 1 (right), the intervention brings minimal impact during motion-consistent phases like "move forward", during motion-transition phases like "locate base" and "assemble them", the switching leads to noticeable degradation. It suggests that the vision modality of the vision-proprioception policy fails to take effect during motion-transition phases.

We further investigate the underlying cause from an optimization perspective. During motion-transition phases, visual cues tend to be subtle and may only differ at the pixel level Tsagkas et al. (2025). As a result, the vision-proprioception policy naturally gravitates toward the more concise proprioceptive signals to minimize the training loss, thereby dominating the optimization Huang et al. (2022); Fan et al. (2023). This dominance suppresses the learning of the vision modality and ultimately leads to under-utilized visual information during motion-transition phases.

To alleviate this, we propose the Gradient Adjustment with Phase-guidance (GAP) algorithm that adaptively modulates the optimization of proprioception, enabling dynamic collaboration between vision and proprioception. Specifically, we first define the motion of the robot using the concise proprioception signals and segment the trajectory into motion-consistent phases. Motion of the robot transits within the intervals between these phases, we thus employ an temporal network like LSTM to model transition processes and estimate the probability that each timestep belongs to motion-transition phases. During policy learning, we guide the vision-proprioception policy to focus on essential visual cues of motion-transition phases, by applying fine-grained gradient adjustment that reduces the magnitude of proprioception's gradient based on estimated probabilities.

Our GAP algorithm facilitates the vision-proprioception policy to effectively utilize proprioception without suppressing the learning of visual modality. GAP is compatible with both conventional and Vision-Language-Action models, and its versatility and effectiveness have been validated by extensive experiments in both simulated and real-world environments. The evaluations cover a wide range of manipulation tasks and includes one-arm and dual-arm robotic setups. We believe this work can offer valuable insights into the development of vision-proprioception policies for robotic manipulation.

## 2 Related Work

**Vision-Proprioception Policy in Manipulation.** Vision has been the most commonly used modality in robotic manipulation policies Zitkovich et al. (2023); Kim et al. (2024); Zeng et al. (2024). While it provides sufficient information to complete many manipulation tasks, visual data often includes a large amount of noise, such as irrelevant background distractions Tsagkas et al. (2025). Therefore, more concise proprioceptive information has been introduced by many works to assist robotic manipulation policy, with the expectation that it can provide complementary and physically grounded information for precise and robust task execution Cong et al. (2022); Mandlekar et al. (2022); Chi et al. (2023); Fu et al. (2024); Wang et al. (2024); Liu et al. (2024). However, existing studies have reported confused observations: some works demonstrate clear improvements when integrating proprioceptive information with vision Cong et al. (2022); Wang et al. (2024), others observe limited gains or even detrimental effects Mandlekar et al. (2022); Octo Model Team et al. (2024). Fu et al. (2024) attributes this to overfitting while Octo Model Team et al. (2024) suggests it arises from causal confusion between the proprioceptive information and the target actions. In this study, we further explore when vision-proprioception policy would fail and introduce a modality-temporality perspective to offer valuable insights into the development of vision-proprioception policies for robotic manipulation.

**Modality Temporality.** In manipulation tasks, each modality's contribution to decision-making can vary significantly over time. For example, in "pick-place" task, policy must first rely on vision to locate the target object. When moving toward the object, proprioception becomes more critical for executing consistent and precise actions. It is proven by strong correlations between variations in modality data and task stages Lee et al. (2019); He et al. (2025); Jiang et al. (2025). Feng et al. (2024) summarizes such property of manipulation tasks as modality temporality. Given this nature of robotic manipulation tasks, recent works have proposed approaches based on dynamic fusion Li et al. (2023); Feng et al. (2024); He et al. (2025) and modality selection Jiang et al. (2025) to improve the performance of multimodal manipulation policies. In this study, we introduce the modality-temporality perspective to understand the roles of vision and propriocetion and propose the gradient adjustment algorithm to enhance dynamic collaboration within the vision-proprioception policy.

## 3 When Would Vision-Proprioception Policy Fail?

In this section, we first formalize the problem and further analyze when vision-proprioception policy would fail from an optimization perspective. The vision-proprioception policy is learned under the Behavior Cloning (BC) paradigm, which can be formulated as the Markov Decision Process (MDP) framework Torabi et al. (2018). Formally, the policy $\pi$ takes the environment observation $o_t \in O$ as input at each timestep $t$. In this work, $o_t$ includes RGB-sensor readings $v_t$, and for vision-proprioception policy $\pi_{v+s}$, it includes robot proprioceptive information $s_t$ additionally. This proprioceptive information consists of the 6D pose of robot's gripper $(p_t^x, p_t^y, p_t^z, \theta_t^x, \theta_t^y, \theta_t^z) \in \mathbb{R}^6$ in Cartesian space and orientation, and a continuous value $g_t \in [0, 1]$ representing the degree of gripper opening, with $g_t = 1$ denoting fully open and $g_t = 0$ denoting fully closed.

The policy $\pi$ maps the observation history to a sequence of actions: $\hat{a}_{t+L} = \pi(o_{t-H:t})$, where $L$ and $H$ indicate the length of predicted action sequence and observation history respectively. For simplicity, we set omit them in the following discussion. The training objective can be formulated as:

$$\pi^* = \operatorname{argmin}_\pi \mathbb{E}_{(o_t, a_t) \sim \tau_e}[\mathcal{L}_{BC}(\pi(o_t)), a_t], \tag{1}$$

where $\tau_e$ is expert demonstration dataset and $a_t$ is action labels. In vanilla BC, $\mathcal{L}$ typically represents the Mean Squared Error (MSE) loss for continuous action spaces, or Cross-Entropy (CE) loss for discrete action spaces. We focus solely on the vanilla MSE loss here.

In this work, we adopt standard joint-learning architecture to design the vision-proprioception policy, which extracts features from both vision and proprioception modalities using two separate chunks $\phi_v, \phi_s$. These features from two modalities are then concatenated and fed into the policy head $\psi$. Although some recent works have tried exploring alternative modality fusion approaches Wang et al. (2024); Feng et al. (2024), concatenation remains the most widely used approach Levine et al. (2016); Cong et al. (2022); Mandlekar et al. (2022). To support our analyze under this fusion approach, we split the first layer of MLP-based policy head $\psi$ into $\psi_s, \psi_v$ and rewrite the action prediction as:

$$\hat{a} = (\psi_s(f_s) + \psi_v(f_v)) \cdot W_{share} + b, \tag{2}$$

where $f_s, f_v$ is the feature extracted by $\phi_s(o), \phi_v(o)$ respectively. Under Gradient Descent (GD)-based policy learning, the optimization of the vision chunk's parameters $\omega_v$ is influenced by:

$$\frac{\partial \mathcal{L}_{BC}}{\partial \omega_v} = \frac{\partial ||\hat{a} - a||_2^2}{\partial \hat{a}} \cdot \frac{\partial (\psi_s(f_s) + \psi_v(f_v)) \cdot W_{share} + b)}{\partial f_v} \cdot \frac{\partial f_v}{\partial \omega_v}. \tag{3}$$

Within the execution trajectory of the task, changes in visual cues are usually subtle compared to proprioceptive signals. For example, when the gripper is closing, visual cues differ only at pixel-level Tsagkas et al. (2025), while concise and low-dimension proprioceptive signals directly represent this process via changes in opening degree $g$. As a result, the vision-proprioception policy naturally gravitates toward proprioceptive signals to minimize the training loss. It leads to optimization dominated by proprioception and suppresses the learning of $\omega_v$ due to vision modality's low contribution to action prediction Huang et al. (2022); Fan et al. (2023).

As shown in Figure 1 (right), such overreliance to proprioception brings negligible impact during motion-consistent phases, since the execution of ongoing movements benefits significantly from proprioceptive signals. However, the initial positions of the target objects vary during testing and the proprioceptive signal does not contain object-related information. During motion-transition phases, the policy is required to accurately locate the target objects. The suppressed learning of vision modality thus regretfully impairs generalization of the vision-proprioception policy.

## 4    Method

To alleviate the suppression of the learning of vision modality during motion-transition phases, we propose the Gradient Adjustment with Phase-guidance (GAP) algorithm. As shown in Figure 2, we initially define the representation of robot's motion and identify motion-consistent phases. Motion-transition phase indicators are then predicted to estimate the probability that each timestep belongs to motion-transition phases. Based on these indicators, we apply fine-grained gradient adjustment during policy learning, facilitating dynamic collaboration within the vision-proprioception policy.

### 4.1    Motion Representation of Robot

Proprioceptive signals of the trajectory $[s_1, s_2, ..., s_N]$ directly provide the state of the gripper's position $p$, orientation $\theta$, and opening degree $g$. The variations in them effectively capture the motion of the robot arm over time. We first define the representation of motion for further motion-transition phase estimation. Specifically, the motion between timestep $i$ and timestep $j$ is defined as: $m_{i:j} = \{p_{i:j}, \theta_{i:j}, g_{i:j}\}$, where $p_{i:j} = p_j - p_i$ denotes the change in the gripper's 3D position, $\theta_{i:j} = \theta_j - \theta_i$ denotes the change in orientation, and $g_{i:j} = g_j - g_i$ denotes the change in gripper opening. Together, these three dimensions provide a complete representation of the robot's motion.

### 4.2    Motion-Transition Phase Estimation

The represented motion captures the movement of robot arm, allowing expert demonstrations to be segmented into sequences of continuous states that correspond to semantically similar motions. To leverage this property for identifying motion-consistent phases, we employ the simple yet effective Change Point Detection (CPD) algorithm Liu et al. (2013); Aminikhanghahi and Cook (2017). The overall motion of a trajectory phase $\tau_{t_1:t_2}$ can be characterized by $m_{t_1:t_2}$. Based on whether the directions of these changes are consistent, we define the following distance between phase motion $m_{t_1:t_2}$ and adjacent motion $m_{i:i+1}$:

$$d(m_{t_1:t_2}, m_{i:i+1}) = -\cos(p_{t_1:t_2}, p_{i:i+1}) - \alpha\cos(\theta_{t_1:t_2}, \theta_{i:i+1}) - \beta(\text{sgn}(g_{t_1:t_2}) == \text{sgn}(g_{i:i+1})), \tag{4}$$

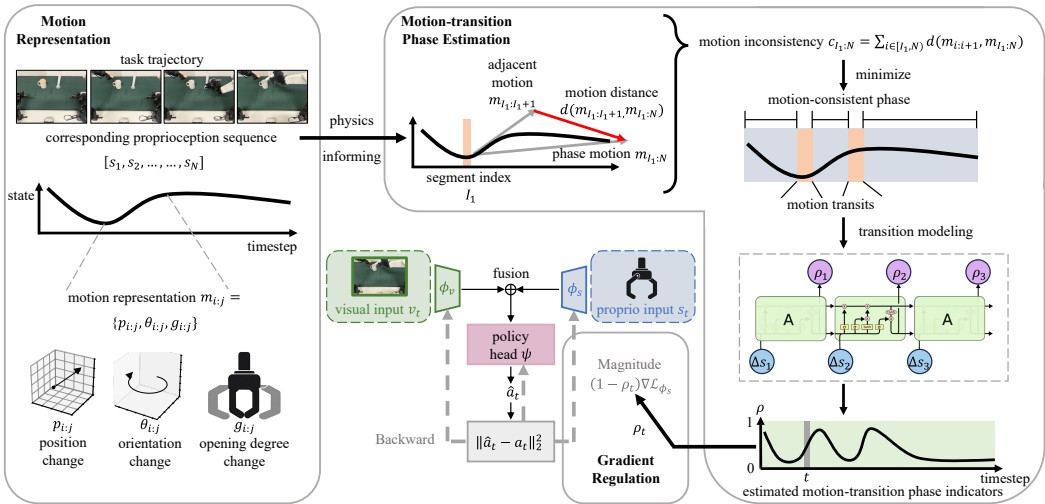

Figure 2: The pipeline of our Gradient Adjustment with Phase-guidance (GAP) algorithm. We define the motion representation and identify the motion-consistent phases by minimizing the total cost between phase motion and each adjacent motion. Motion-transition phase indicators are then estimated to reduce the magnitude of proprioception's backward gradient. GAP facilitates vision-proprioception policies to effectively utilize proprioception without suppressing vision modality.

where $\text{sgn}(\cdot)$ denotes the sign function, and $\alpha, \beta$ are weighting factors for the orientation and opening degree component respectively. The statistic that measure the motion inconsistency of phase $\tau_{t_1:t_2}$ is defined as $c_{t_1:t_2} = \sum_{i=t_1}^{t_2-1} d(m_{t_1:t_2}, m_{i:i+1})$. The Change Point Detection algorithm leverages dynamic programming to identify a set of indices $I$ that minimize the total cost $\sum_I c_{\tau_I}$, segmenting the trajectory into motion-consistent phases.

Motion of the robot transits within the intervals between these phases, requiring the policy to locate target object. Vision is therefore expected to play a more significant role. To model motion transitions, we further utilize the temporal differences of proprioceptive information $\Delta s_i = s_{i+1} - s_i$ and leverage their sequential context with an temporal network such as LSTM. It predicts motion-transition phase indicators $\rho_i$ to estimate the probability that timestep $i$ belongs to motion-transition phases. The predicted indicators $\rho$ is under the supervision of indices set $I$. Additionally, for timesteps within a range near the transition, we reduce the penalty applied to them in order to better capture the inherently continuous and gradual transition process.

## 4.3 Gradient adjustment for Modality Collaboration

The vision-proprioception policy extracts features from both vision and proprioception modalities using two separate chunks $\phi_v, \phi_s$, which consist of an encoder and a temporal transformer, these features are then fused and fed into policy head to predict the action. However, since visual cues during motion-transition phases may be subtle, the policy tends to rely heavily on features of proprioception. As a result, the gradient optimization for corresponding samples becomes dominated by proprioceptive inputs, which in turn constrains the learning of the vision modality chunk $\phi_v$.

To mitigate this, we employ gradient adjustment to control the optimization of proprioceptive chunk $\phi_s$ during motion-transition phases, thereby guiding the vision-proprioception policy to focus more on visual cues and preventing the degradation of its generalization. Concretely, in the $j$-th epoch of Gradient Descent (GD)-based optimization, the parameters of the proprioceptive feature chunk $\omega_s^j$ are updated according to the following formula:

$$\omega_s^{j+1} = \omega_s^j - \lambda \cdot (1 - \rho) \cdot \eta \nabla \omega_s^j \mathcal{L}_{BC}(\omega_s^j), \tag{5}$$

where $\eta$ is the learning rate, $\lambda$ is a hyper-parameter that controls the degree of adjustment. For each timestep, we modulate the magnitude of the proprioception backward gradient based on its indicator $\rho$ of belonging to motion-transition phases. The higher value of $\rho$ leads to greater degree of modulation.

By applying gradient adjustment with phase-guidance as illustrated in Algorithm 1, the vision-proprioception policy is enabled to effectively leverage proprioceptive information without compromising its generalization ability.

---

**Algorithm 1** Vision-Proprioception Policy Learning with Gradient Adjustment

---

**Notations:** Expert demonstrations $o_e$, proprioceptive signals $s_e$, epoch number $T$, vision-proprioception policy $\pi_{v+s}$, proprioception chunk parameters $\omega_s$, vision chunk parameters $\omega_v$.

**Motion-Transition Phase Estimation**
Identify motion-consistent phases by Change Point Detection $I \leftarrow \text{CPD}(s_e)$;
Predict motion-transition phase indicators $\rho \leftarrow \text{LSTM}(\Delta s_e)$ ;

**Gradient Adjustment during Policy Learning**
**for** $j = 0, 1, \cdots, T - 1$ **do**
    Sample a fresh mini-batch $B_j$ from expert demonstrations $o_e$;
    Feed-forward the batched data $B_j$ to $\pi_{v+s}$;
    Calculate average indicator $\rho_j$ of $B_j$;
    Update proprioception chunk $\omega_s^{j+1}$ using Equation 5;
    Update vision chunk $\omega_v^{j+1}$.
**end for**

---

# 5   Experiments

In this section, we conduct validate the versatility and effectiveness of our proposed Gradient Adjustment with Phase-guidance (GAP) algorithm through a series of question-driven experiments. The evaluations comprehensively cover a wide range of manipulation tasks, including simple pick-and-place tasks, rotation-sensitive tasks, as well as long-horizon and contact-rich tasks.

## 5.1   Experimental Setup

We select two simulated environments as our benchmarks: MetaWorld Yu et al. (2020) and RoboSuite Zhu et al. (2020). Tasks in MetaWorld are relatively simple, featuring a 4-dimensional action space that includes the gripper's position and its opening degree, while tasks in RoboSuite involve complex scenarios, longer task sequence horizons and richer physical interactions, with the action space further including the orientation of the gripper. For real-world experiments shown in Figure 3, we use a 6-DoF xArm 6 robotic arm equipped with a Robotiq gripper for all one-arm tasks. Moreover, we utilize the open-source Cobot Magic platform to support tasks that require dual-arm collaboration. In all tasks, the initial position of target object varies randomly in each validation, while the initial position of gripper remains fixed.

## 5.2   Can vision-proprioception policies outperform after GAP?

Vision-Proprioception policies perform generally worse than vision-only policies. Can they outperform vision-only policies after our GAP algorithm is applied? To answer this, we conducted comparative analyses between our algorithm and the following baselines:

- MS-Bot Feng et al. (2024): this method uses state tokens with stage information to guide the dynamic collaboration of modalities within multi-modality policy.

- Auxiliary Loss (Aux): following HumanPlus Fu et al. (2024), we use visual feature to predict the next frames as an auxiliary loss, which tries to enhance the vision modality.

- Mask: to prevents the overfitting to specific modality, RDT-1B Liu et al. (2024) randomly and independently masks each uni-modal input with a certain probability during encoding. We adapt the algorithm by masking only proprioception modality instead.

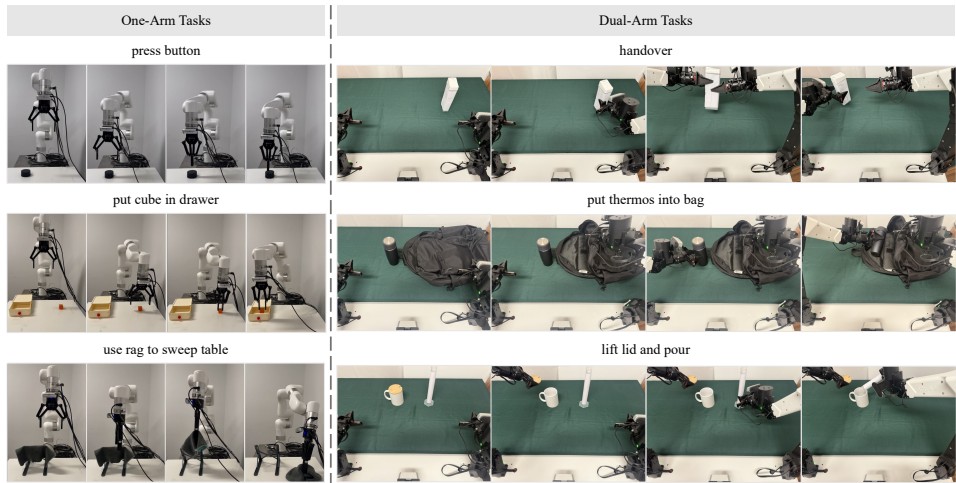

Figure 3: Visualization of real-world tasks. Our experiments cover a wide range of manipulation tasks, including both One-Arm and Dual-Arm Setups.

Results in Table 1 demonstrate that vision-proprioception policies with our GAP applied outperform vision-only policies and other methods. Although MS-Bot achieves overall improvements over the vision-only policy by incorporating stage information, it focuses on the semantic stage instead of motion-transition phases. As a result, its benefits are marginal in tasks like "push-wall" and "lift lid and pour", where motion frequently transits. This highlights the necessity of fine-grained gradient adjustment during motion-transition phases. Auxiliary loss forces the vision-proprioception policy to concentrate on visual input during the whole task, which falls short in tasks requiring proprioception to enhance the precision and robustness of manipulation, such as "threading". Meanwhile, masking the proprioceptive input with a fixed probability overlooks the modality temporality of manipulation tasks, resulting in minimal improvement. By adaptively applying fine-grained gradient adjustment during motion-transition phases, GAP enables the vision-proprioception policy to effectively leverage these two modalities and outperform both the vision-only policy and other methods.

Table 1: Comparisons with other methods in both simulated and real-world environments. The vision-proprioception policies after our gradient adjustment significantly outperform other methods.

| Suite | Meta-World | | | | | RoboSuite | | |
|---|---|---|---|---|---|---|---|---|
| Method \ Task | pick-place | assembly | disassemble | push-wall | bin-picking | hammer | stack | threading |
| Vision-only | 92% | 82% | 85% | 64% | 63% | 86% | 67% | 44% |
| Concatenation | 79% | 76% | 80% | 56% | 49% | 79% | 56% | 34% |
| MS-Bot Feng et al. (2024) | 90% | 93% | 88% | 67% | 70% | 88% | 70% | 51% |
| Aux Fu et al. (2024) | 89% | 93% | 78% | 51% | 55% | 72% | 55% | 47% |
| Mask Liu et al. (2024) | 86% | 90% | 84% | 82% | 61% | 79% | 62% | 48% |
| GAP (Ours) | 94% | 96% | 91% | 73% | 70% | 91% | 77% | 52% |

| Setup | Real One-Arm | | | Real Dual-Arm | | |
|---|---|---|---|---|---|---|
| Method \ Task | press button | put cube in drawer | use rag to sweep table | handover | put thermos into bag | lift lid and pour |
| Vision-only | 18/20 | 14/20 | 9/20 | 15/20 | 11/20 | 9/20 |
| Concatenation | 12/20 | 11/20 | 5/20 | 12/20 | 7/20 | 5/20 |
| MS-Bot Feng et al. (2024) | 20/20 | 16/20 | 11/20 | 16/20 | 13/20 | 10/20 |
| Aux Fu et al. (2024) | 19/20 | 16/20 | 11/20 | 15/20 | 13/20 | 8/20 |
| Mask Liu et al. (2024) | 18/20 | 14/20 | 7/20 | 15/20 | 9/20 | 7/20 |
| GAP (Ours) | 20/20 | 17/20 | 13/20 | 18/20 | 16/20 | 15/20 |

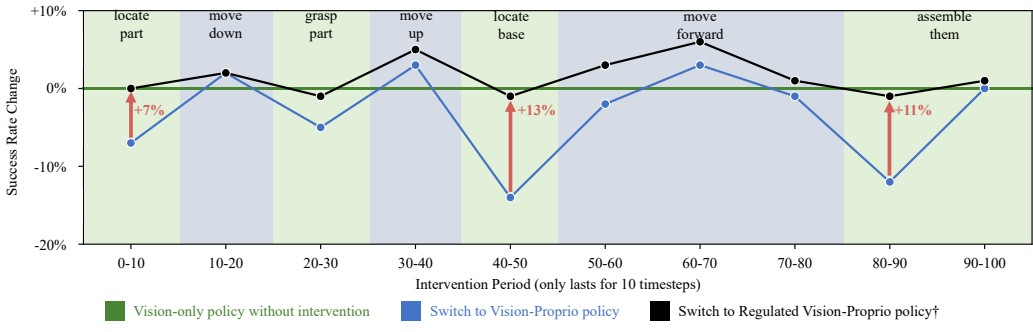

Figure 4: The Intervention experiment we conduct with the regulated vision-proprioception policy. The sight changes in success rate indicate that GAP does enhance the utilization of vision modality.

## 5.3 Does GAP enhance the utilization of vision modality?

Although vision-proprioception policies outperform vision-only policies after apply GAP, it remains unclear whether GAP truly enhances the utilization of the vision modality within vision-proprioception policies. To answer this, we first conducted intervention experiment under the same settings as described in Section 1. As shown in the Figure 4, the degrees of suppression of vision modality during motion-transition phases are significantly reduced after applying GAP, indicating GAP does enhance the utilization of vision modality. We further evaluated the generalization of the vision-proprioception policies in out-of-distribution (OOD) scenarios. In each scenario, the initial distribution of object positions differs from that in the training dataset of expert demonstrations. The vision-only policies are less affected by such changes due to well-utilized vision modality as demonstrated in Tabel 2. Vision-proprioception policies exhibit poor generalization with suppressed vision. Meanwhile, Our algorithm alleviates this by regulating the optimization of the proprioceptive, preventing the suppression. The maintained superior performance over vision-only policy also indicates the effectiveness of introducing proprioception modality for precise and robust manipulation.

Table 2: Experiments under out-of-distribution settings. For each task, our proposed GAP algorithm enhances the generalization of the vision-proprioception policies.

| Setup | Meta-World | | RoboSuite | | Real One-Arm | Real Dual-Arm |
|---|---|---|---|---|---|---|
| Method \ Task | assembly | bin-picking | stack | threading | put cube in drawer | handover |
| Vision-only | 78% | 59% | 63% | 32% | 12/20 | 12/20 |
| Concatenation | 62% | 32% | 49% | 28% | 7/20 | 9/20 |
| Ours | 88% | 67% | 72% | 49% | 15/20 | 15/20 |

## 5.4 Is GAP compatible with Vision-Language-Action models?

Above experiments have demonstrated that our algorithm facilitates dynamic collaboration within conventional vision-proprioception models. We further investigate is GAP compatible with Vision-Language-Action (VLA) models. Specifically, we compare fine-tuned Octo model Octo Model Team et al. (2024) using only visual information (Octo-V) versus using both vision and proprioception (Octo-VP), and tries to apply our gradient adjustment algorithm during fine-tuning. As reported in the original paper, policies trained with additional propioception seemed generally worse than vision-only policies. We observe the same trend across various tasks in Table 3. However, after applying our gradient adjustment algorithm, Octo-VP† achieves an average improvement of 17% and exhibits stronger generalization ability than Octo-V. These results suggest that our algorithm effectively enhances dynamic collaboration between vision and proprioception within VLA models.

Table 3: Performances of fine-tuned Octo. † indicates GAP is applied.

| Suite | | Meta-World | | RoboSuite | |
|---|---|---|---|---|---|
| Model | Task | disassemble | push-wall | hammer | threading |
| Octo-V | | 95% | 77% | 92% | 69% |
| Octo-VP | | 82% | 65% | 88% | 57% |
| Octo-VP† | | 100% | 85% | 97% | 78% |

## 5.5 Can our algorithm be applied to various modality fusion approaches?

The preliminary results in Section 3 reveal that the vision-proprioception policy using straightforward concatenation tends to perform worse than the vision-only policy. We further explore a broader set of fusion approaches and validate the versatility of our algorithm. Specifically, we apply GAP to three typical and widely used fusion approaches: concatenation, summation and FiLM Perez et al. (2018).

As reported in Table 4, vision-only policies outperforms all three fusion approaches in tasks such as "pick-place" and "hammer", indicating that vision modality suffice for certain tasks. However, they fail drastically in task "threading" due to demands for precise manipulation and exhibits suboptimal performance in task "push-wall", which involves visual occlusions at the target location, highlighting the necessity of the inclusion of proprioceptive information for precise and robust manipulation.

Table 4: Performance of typical fusion approaches combined with GAP. † indicates GAP is applied.

| Suite | | Meta-World | | | | | RoboSuite | | |
|---|---|---|---|---|---|---|---|---|---|
| Method | Task | pick-place | assembly | disassemble | push-wall | bin-picking | hammer | stack | threading |
| Vision-only | | 92% | 82% | 85% | 64% | 63% | 86% | 67% | 44% |
| Concatenation | | 79% | 76% | 80% | 56% | 49% | 79% | 56% | 34% |
| Summation | | 78% | 95% | 80% | 54% | 61% | 75% | 49% | 30% |
| FiLM | | 75% | 91% | 47% | 67% | 59% | 76% | 53% | 41% |
| Concatenation† | | 94% | 96% | 91% | 73% | 70% | 91% | 77% | 52% |
| Summation† | | 92% | 97% | 93% | 66% | 70% | 88% | 82% | 48% |
| FiLM† | | 90% | 94% | 85% | 74% | 68% | 95% | 72% | 46% |

Concatenation preserves raw features from both modalities, but the high-dimensional redundancy hinders the policy to dynamically utilize each modality. As a result, it underperforms in tasks like "push-wall", where effective coordination is required. Simple summation may obscure critical details, whose limitation is evident in precise manipulation tasks such as "threading" and "push-wall". Meanwhile, FiLM applies affine transformations to conditionally adjust features, making it more suitable for tasks requiring modality collaboration. For instance, it achieves a notably higher score in "push-wall" task. However, its performance tends to degrade in simpler tasks where such complex conditioning may be unnecessary. Conversely, GAP successfully unlocked the full potential of the vision-proprioception policy, outperforming vision-only policies in all three fusion approaches.

## 6 Conclusion and Limitation

In this study, we illustrate that vision-proprioception policy would fail during motion-transition phases due to its suppressed vision modality. To alleviate this, we propose the Gradient Adjustment with Phase-guidance (GAP) algorithm, enabling dynamic collaboration between vision and proprioception within vision-proprioception policy. We believe this work can offer valuable insights into the development of vision-proprioception policies for robotic manipulation.

**Limitations.** All vision-proprioception policies are trained on single embodiment in this work. As existing large-scale datasets often contain diverse embodiments, exploring the role of proprioception in cross-embodiment datasets would be promising for future research.

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
