# OpenReview forum: "When Would Vision-Proprioception Policy Fail in Robotic Manipulation?"
_NeurIPS.cc/2025/Conference — Submitted to NeurIPS 2025_

### Official Review · Reviewer_48PM · 2025-06-29

**Clarity:** 4
**Significance:** 4
**Originality:** 3
**Rating:** 5
**Confidence:** 4

**Summary:**

This paper investigates the failure of imitation learning policies that leverage both vision and proprioception information as inputs. Empirical results demonstrate that policies conditioned on both modalities tend to fail in motion-transition phases where the policy has to pay attention to detailed visual cues. Motivated by this result, the authors propose the Gradient Adjustment with Phase-guidance (GAP) framework. GAP first identifies motion-transition phases in demonstration trajectories using Change Point Detection (CPD) and an LSTM. The learned LSTM outputs a continuous scalar score that predicts how likely the robot is in a motion-transition phase. Based on this prediction, the magnitude of the gradient of the imitation learning loss with respect to the proprioceptive inputs is modulated. This modulation encourages the policy to pay more close attention to visual cues in motion-transition phases. Experiments in both simulation and the real-world demonstrate the effectiveness of GAP.

**Questions:**

* It remains a bit unclear how CPD and LSTM are leveraged together to perform motion-transition phase estimation. Specifically:

  * What is the target output for the LSTM training? Is it a binary 0-1 label indicating the motion-consistent and motion-transition phases as computed by the CPD algorithm?
  * Could we possibly eliminate the LSTM training phase, for instance by simply smoothing the motion phase outputs of CPD? The versatility of GAP would improve if it did not require the additional LSTM training phase.

* I wonder whether the ignorance of vision inputs is still an issue for models like Open-VLA, where strong general-purpose vision/language features such as DinoV2 and SigLIP are employed. Some comments on this point would be appreciated.

**Ethical Concerns:**

["NO or VERY MINOR ethics concerns only"]

**Final Justification:**

The rebuttal by authors resolved all the major concerns and the questions that I raised in my original review. The additional experiments the authors have conducted have improved the clarify and the significance of the paper. Thus I have updated those scores accordingly.

**Limitations:**

yes

**Quality:**

3

**Strengths And Weaknesses:**

**Strengths:**

* The paper first conducts an investigative study on when vision-proprio policies tend to be more vulnerable than vision-only policies to form a scientific hypothesis. This hypothesis leads to the design of GAP, which is shown to be effective through a series of experiments. Such a hypothesis-driven approach to robot learning research is highly appreciative.

* The proposed GAP framework seems versatile and applicable to various imitation learning architectures (e.g., MLP-based, diffusion, flow-matching, etc.).

* The paper conducts thorough experiments, including ablation of multiple policy architectures and fusion techniques across multiple tasks in both simulation and the real-world.


**Weaknesses:**

* Lack of key information in the experiments: I was unable to identify the number of rollouts per policy per task in the Meta-World and RoboSuite simulation benchmarks. Lack of sample size makes it impossible to judge the significance of empirical results. (For instance, 90% vs. 96% success rates may be insignificant under 50 rollouts but are likely significant with 500 rollouts, due to smaller statistical noise.)

* The OOD experiment in Section 5.3 is a bit contrived, since the authors only test GAP under different object placements. More visual distribution shifts such as changes in background color, object color, or object shape might lead to poorer performance of GAP, since GAP-trained policies rely less on proprioceptive information that is unaffected by those visual distribution shifts.

---

> ### Author Rebuttal · Authors · 2025-07-31
>
> We sincerely appreciate the reviewer's recognition of our work and the thoughtful feedback. We have carefully addressed each of your concerns and questions in detail below.
>
> $\color{blue}Question \space 1: $ The lack of key information of experiment settings.
>
> $\color{darkgreen}Response \space 1: $
> We acknowledge this important oversight and provide the missing experimental details: **each task of our experiments in Meta-World and RoboSuite is evaluated with 100 rollouts per policy per task.** These relatively large rollouts ensure the significance of our results, which is further supported by the **multiple seeds results provided in Section B.1 of the supplementary materials**. Our experiments include 5 different random seeds, and the stable and consistently higher results across seeds demonstrate the significance and robustness of our method.
>
> We apologize for the oversight in not clearly reporting these crucial experimental details. We will ensure that all experimental settings, including the number of rollouts and random seeds, are explicitly stated in the revised version to provide readers with complete information for evaluating the significance of our empirical results.
>
> $\color{blue}Question \space 2: $ Limited Out-of-Distribution evaluation.
>
> $\color{darkgreen}Response \space 2: $
> We sincerely appreciate the reviewer's valuable feedback regarding our out-of-distribution (OOD) experiments. The reviewer correctly points out that our initial OOD evaluation in Section 5.3 was limited to different object placements, and suggests that more comprehensive visual distribution shifts would provide a more thorough assessment of our method's robustness.
>
> Following the reviewer's insightful suggestion, we conducted additional comprehensive OOD experiments to thoroughly evaluate GAP's robustness under various visual distribution shifts:
>
> - **Table Color**: We modified the table color during testing to create visual distribution shifts that affect the overall scene appearance.
>
> - **Object Color**: We tested with objects of different colors than those used during training, introducing color-based visual perturbations.
>
> - **Lighting**: We introduced colored spot lighting during testing to create additional visual disturbances.
>
> **Table 1: Enhanced OOD Evaluation of Task "assembly" in Meta-World**
> | Method | Placement | Table Color | Object Color | Lighting |
> |:------:|:--------:|:------------:|:-------------:|:---------:|
> | **Vision-only** | 78% | 80% | 75% | 73% |
> | **Concatenation** | 62% | 72% | 76% | 74% |
> | **GAP (Ours)** | **88%** | **85%** | **83%** | **90%** |
>
> **Table 2: Enhanced OOD Evaluation of Task "put cube in drawer" in Real World**
> | Method | Placement | Table Color | Object Color | Lighting |
> |:------:|:--------:|:------------:|:-------------:|:---------:|
> | **Vision-only** | 60% | 65% | 60% | 55% |
> | **Concatenation** | 35% | 50% | 55% | 50% |
> | **GAP (Ours)** | **75%** | **80%** | **85%** | **80%** |
>
> The results from Tables 1 and 2 demonstrate that GAP maintains superior performance compared to baseline methods across all challenging OOD scenarios.
>
> Across all OOD settings, **placement changes have the most significant impact on performance, confirming that accurate perception of target object positions is fundamental for successful manipulation tasks**. While concatenation-based methods that rely heavily on proprioceptive information (which remains unaffected by visual disturbances) may maintain relatively stable performance across different visual perturbations, their overall effectiveness is suboptimal due to the lack of precise position perception.
>
> GAP's superior performance compared to vision-only approaches demonstrates that introducing proprioceptive information to collaborate with visual information can enhance policy performance in OOD scenarios. Our results further validate the importance of our work in advancing research on visual-proprioceptive collaboration.
>
> We thank the reviewer for this constructive feedback, which has led to a more comprehensive evaluation.
>
> $\color{blue}Question \space 3: $ Roles of LSTM in Motion-Transition Phase Estimation module. (its output and comparisons with smoothing)
>
> $\color{darkgreen}Response \space 3: $
> We appreciate the reviewer's question regarding the role of LSTM in our motion-transition phase estimation module. We acknowledge that our brief introduction to the LSTM component may have caused some confusion, and we will provide a detailed explanation of its function along with experimental validation of its effectiveness.
>
> Our motion-transition phase estimation module employs a two-stage design strategy: first identifying critical transition moments through Change Point Detection (CPD) algorithm, then utilizing LSTM networks to refine these discrete detection results.
>
> **Technical Design Rationale:**
> The CPD algorithm outputs discrete transition point indices (e.g., [I_1=10, I_2=25, I_3=53, ...]), which mark the timesteps where significant changes in robot motion states occur. However, actual motion transitions are gradual processes, and **the discrete outputs from CPD cannot accurately capture the continuity and uncertainty of transition processes**. For instance, when CPD detects timestep 25 as a transition point, adjacent timesteps 23-27 may all be part of the transition process, but CPD cannot quantify the transition degree at these timesteps.
>
> **LSTM's Critical Function:**
> The LSTM network learns temporal dependencies to transform CPD's discrete detection results into continuous transition probability distributions. Specifically, **the LSTM outputs probability values (0-1 range) for each timestep belonging to transition phases**, which enables more precise description of transition process graduality and reduction of potential detection errors introduced by CPD algorithm.
>
> **Experimental Validation:**
> To verify the necessity of the LSTM component, we designed comparative experiments comparing LSTM-based methods with the following alternatives:
>
> - **Fixed Adjustment Strategy:** Apply fixed gradient adjustment magnitude ρ to transition points detected by CPD
> - **Smooth:** Following KOI [1], treat CPD output as the mean of a Gaussian distribution for smoothing
>
> **Table 3: Ablation Studies on the LSTM component**
> | Method | hammer | threading | put cube in drawer |
> |:------:|:------:|:---------:|:------------------:|
> | **Vision-only** | 86% | 44% | 70% |
> | **Fixed ρ=0.3** | 77% | 44% | 65% |
> | **Fixed ρ=0.5** | 89% | 48% | 75% |
> | **Fixed ρ=0.7** | 82% | 38% | 65% |
> | **Smooth** | 88% | 46% | 80% |
> | **GAP (Ours)** | **91%** | **52%** | **85%** |
>
> The experimental results indicate that although alternative methods are capable of modeling the transition process to a certain degree, the LSTM-based approach consistently achieves superior performance.
>
> **Robustness Verification:**
> To further validate LSTM's error tolerance capability for CPD detection errors, we conducted noise injection experiments: randomly selecting directions with 0.5 probability and recursively shifting each CPD output index. As shown in Table 4, even with significant noise injected into CPD outputs, policy performance remains robust, proving the effectiveness of LSTM's continuous modeling.
>
> **Table 4: Noise Injection Experiments**
> | Method | hammer | threading | put cube in drawer |
> |:------:|:------:|:---------:|:------------------:|
> | **Vision-only** | 86% | 44% | 70% |
> | **Concatenation** | 79% | 34% | 55% |
> | **Noise-Injected** | 88% | 49% | 85% |
> | **GAP (Ours)** | **91%** | **52%** | **85%** |
>
> We appreciate the reviewer's valuable suggestions. In the revised version, we will provide more detailed technical explanations and experimental analyses to help readers better understand the design philosophy and implementation details of this module.
>
> $\color{blue}Question \space 4: $ Strong general-purpose vision features in Vision-Proprioception policies.
>
> $\color{darkgreen}Response \space 4: $
> We sincerely appreciate the reviewer's valuable question. We have conducted in-depth analysis and additional experiments to address it.
>
> As the reviewer correctly points out, vision foundation models such as DinoV2 and SigLIP are primarily trained on web data to obtain general-purpose features. Therefore, in robotics manipulation, these models often require fine-tuning on specific robot scenarios. Both the reported observations from Octo and our results in Section 5.4 demonstrate that the challenges we address exist regardless of whether the models are pre-trained on large-scale robot data or fine-tuned on smaller datasets.
>
> To further validate our approach, we conducted experiments on OpenVLA-OFT [2], utilizing its strong general-purpose vision features from SigLIP and DINOv2 as the foundation and fine-tuning OpenVLA with different inputs.
>
> **Table 5: Vision-Proprioception Policy with Strong General-Purpose Vision Features**
> | | assembly | threading | use rag to sweep table |
> |:------:|:--------:|:---------:|:-------------------:|
> | **Vision-only** | 91% | 59% | 65% |
> | **With proprio** | 95% | 62% | 75% |
> | **With GAP** | **98%** | **77%** | **95%** |
>
> The results in Table 5 demonstrate that when strong general-purpose vision features are employed, introducing proprioceptive information may not directly degrade performances compared to vision-only. Nevertheless, applying our GAP method to improve the fine-tuning process leads to better performance. These results indicate that even when utilizing strong general-purpose vision features, promoting effective visual-proprioceptive collaboration remains a meaningful research problem.
>
> ---
>
> [1] Lu J, Xia W, Wang D, et al. Koi: Accelerating online imitation learning via hybrid key-state guidance. CoRL, 2024.
>
> [2] Kim M J, Finn C, Liang P. Fine-tuning vision-language-action models: Optimizing speed and success. arXiv preprint arXiv:2502.19645, 2025.

---

> > ### Comment · Reviewer_48PM · 2025-08-04
> > **Acknowledgement of Rebuttal**
> >
> > I sincerely thank the authors for their thorough response. It resolved all the major concerns and the questions that I raised in my original review. The additional experiments the authors have conducted will improve the clarify and the significance of the paper. Thus I have updated those scores accordingly.

---

> ### Author Response · Authors · 2025-08-04
>
> Dear reviewer, your perspective on whether our replies have adequately addressed your concerns would be valuable to us. We would be delighted to provide any additional details that might be helpful.

---

> ### Author Response · Authors · 2025-08-04
> **Thank you for your response**
>
> Thank you for your valuable feedback and suggestions. We will incorporate the additional results and experiments into our revised version.

---

### Official Review · Reviewer_RuJT · 2025-07-01

**Clarity:** 4
**Significance:** 4
**Originality:** 3
**Rating:** 5
**Confidence:** 4

**Summary:**

This paper examines the impact of proprioceptive measurements in visual-proprioceptive policy learning. The paper hypothesizes that the gradients from the proprioceptive signals dominate the optimization process during motion transition phases, where the visual cues are less prominent, leading to under-utilized visual information. This hypothesis is supported by several experiments in both real-world and simulation settings. The paper further proposes Gradient Adjustment with Phase-guidance (GAP) to alleviate such an issue. The experimental results show that GAP greatly improves the success rate of visual-proprioceptive policies.

**Questions:**

1. Motion transition phases can be one of the causes for gradient imbalance. Do you think it is possible to extend the proposed method to a more general framework, where the system automatically figures out when to put attention on proprioceptive signals or visual cues?

**Ethical Concerns:**

["NO or VERY MINOR ethics concerns only"]

**Final Justification:**

I thank the authors for their efforts in providing the rebuttal. Most of my concerns have been addressed. However, the heuristic design of the CPD algorithm still remains a weak point of the paper, although empirically it's shown to provide better results. In particular, I'm curious about the definition of a motion transition phase. The authors mentioned that human annotation performs worse than the CPD because humans tend to favor semantically annotated results. But without a definition of a ground truth, how can one be sure this is the case? Secondly, without a clear definition of the motion transition phase, how can one design a principled approach (compared to heuristic CPD) to address the gradient imbalance issue?

Nevertheless, this paper offers a new perspective on why robot policy performance may decline with the inclusion of proprioceptive signals. In addition, it proposes a solution to mitigate such an effect. As a result, I will maintain my rating.

**Limitations:**

1. The paper primarily focuses on methods with two detached heads for visual and proprioceptive inputs. It is unclear if the same results can be obtained from systems with a shared encoder.
2. As mentioned in the paper, all the visual-proprioceptive policies are trained on single embodiment. There is no guarantee that the results can be extrapolated to methods trained on cross-embodiment datasets.

**Paper Formatting Concerns:**

No formatting concerns.

**Quality:**

3

**Strengths And Weaknesses:**

Strengths

1. The paper shows that visual-proprioceptive policy can suffer from performance degradation when visual cues are subtle compared to proprioceptive signals. It further demonstrates that such an issue is caused by imbalanced gradient flows. In my opinion, this finding provides great insights into training visual-proprioceptive policies.

2. The proposed GAP method is simple, yet it is shown to effectively improve the success rate of visual-proprioceptive policies.

3. The paper presents 4 different experiments to help understand the effect of GAP. The experiments show that:
a) GAP can improve the performance of visual-proprioceptive policies.
b) GAP enhances the focus on visual inputs, allowing better generalization to different object locations.
c) GAP is compatible with vision-language-model Octo-VP.
d) GAP can be applied to different fusion approaches, and in general, improve the performance.

Weaknesses
1. The description of the motion transition phase in the introduction is intuitive. However, the definition is unclear. An explanation of why the motion transition phase can be captured by the Change Point Detection with heuristic rules is important for readers to understand the paper.
2. In addition to the above point, evaluations on how well the CPD algorithm can capture the motion transition phase can greatly improve the paper.
3. The proposed GAP method depends on a trained LSTM to predict the motion transition phase. However, there is no evidence on how well the network performs, as well as it’s effect on the overall performance.

---

> ### Author Rebuttal · Authors · 2025-07-31
>
> We are truly grateful for the reviewer's positive evaluation of our work and valuable suggestions. Below, we provide detailed responses to each of your concerns and questions.
>
> $\color{blue}Question \space 1: $ The lack of explanations of Motion-Transition Estimation module.
>
> $\color{darkgreen}Response \space 1: $
> We sincerely appreciate the reviewer's valuable feedback regarding the clarity of our motion transition phase definition. We acknowledge that providing more detailed explanations will significantly enhance readers' understanding of our approach.
>
> Our design is motivated by intervention experiments that revealed a critical observation: during phases such as "locate base" and "assemble them," the visual modality does not play its expected role in robot manipulation. Based on this insight, we specifically designed heuristic rules in our Change Point Detection (CPD) algorithm to identify these corresponding phases for subsequent gradient adjustment.
>
> The core of our approach lies in the motion-consistency distance defined in Equation 4:
>
> $$d(m_{t_1:t_2},m_{i:i+1}) = - \text{cos}(p_{t_1:t_2}, p_{i:i+1}) - \alpha \text{cos}(\theta_{t_1:t_2}, \theta_{i:i+1}) - \beta (\mathrm{sgn}(g_{t_1:t_2})==\mathrm{sgn}(g_{i:i+1}))$$
>
> This distance metric is carefully designed to capture different aspects of robot motion. The first term captures spatial position changes of the robot (e.g., moving forward); the following term captures rotational changes of the robot (e.g., rotating for pouring actions); the last term captures gripper opening state changes (e.g., grasping objects).
>
> When the robot transitions from one motion type to another (e.g., from "move forward" to "grasp"), the corresponding distance becomes significantly large. The CPD algorithm then minimizes the sum of distances to avoid grouping trajectory segments involving different motion types into the same phase.
>
> Our experimental results in the main text, along with the ablation studies presented in the following question, demonstrate the effectiveness of this design. The heuristic rules successfully identify motion transition phases, enabling more effective gradient adjustments during policy learning.
>
> We thank the reviewer for this constructive feedback. In the revised version, we will add more detailed explanations to help readers better understand our motion transition phase estimation module.
>
> $\color{blue}Question \space 2: $ The absence of analysis of Motion-Transition Phase Estimation module. (ablation studies on CPD and LSTM)
>
> $\color{darkgreen}Response \space 2: $
> We value your feedback and recognize that providing detailed explanations and ablation studies of this module can help readers better understand our approach and further enhance the quality of our paper.
>
> To thoroughly analyze the effectiveness and robustness of this module, we additionally conducted comprehensive ablation studies. Specifically, we kept all other components unchanged and tested the impact of different trajectory decomposition algorithms on policy learning. The detailed descriptions of the comparison methods are as follows:
>
> - **Human:** Phase transitions are manually labeled by human experts.
>
> - **HDBSCAN:** Following Emma-X [1], we utilize the HDBSCAN algorithm to cluster our distance metric defined in Equation 4, decomposing trajectories into phases.
>
> - **CoTPC [2]:** To decompose task trajectories into temporally close and functionally similar subskills, CoTPC applies the CPD algorithm on the sequence of robot actions using cosine distance.
>
> **Table 1: Performance Comparison between GAP and other decomposition methods**
> | Method | hammer | threading | put cube in drawer |
> |:------:|:------:|:---------:|:------------------:|
> | **Vision-only** | 86% | 44% | 70% |
> | **Human** | 88% | 47% | 75% |
> | **HDBSCAN** | 82% | 37% | 65% |
> | **CoTPC** | 85% | 50% | 85% |
> | **GAP (Ours)** | **91%** | **52%** | **85%** |
>
> As shown in Table 1, human labels provide a slight improvement in performance. Since human annotators tend to decompose trajectories from a semantic perspective rather than based on motion, this approach may miss certain transition points, resulting in only limited gains. The cluster-based HDBSCAN method disrupts the temporal structure inherent to manipulation tasks, leading to even worse policy performance compared to human labeling. Notably, both CoTPC and our GAP framework employ the CPD algorithm for phase estimation and generally outperform human annotation across most tasks, suggesting that automated phase estimation can capture motion transitions more effectively than manual labeling. However, CoTPC relies on a simple cosine distance, which lacks the ability to represent complex robot motions, thus yielding less significant performance improvements compared to GAP. In our approach, the motion-consistent distance defined in Eq 4 captures spatial position changes, rotational changes, and gripper opening state changes, enabling more accurate and efficient detection of motion transitions. **These results highlight the effectiveness of our motion-consistent distance metric and the CPD algorithm in achieving precise phase estimation**.
>
> Additionally, we performed detailed ablation studies on the LSTM component within this module. Specifically, we compared the LSTM-based phase transition modeling with the following alternatives:
> - **Fixed ρ:** For the phase transition points output by CPD, we applied a fixed gradient adjustment magnitude ρ to the corresponding samples during policy learning.
> - **Smooth:** Following KOI [3], we modeled the transition process by treating the CPD output index as the mean of a Gaussian distribution.
>
> **Table 2: Ablation Studies on the LSTM component**
> | Method | hammer | threading | put cube in drawer |
> |:------:|:------:|:---------:|:------------------:|
> | **Vision-only** | 86% | 44% | 70% |
> | **Fixed ρ=0.3** | 77% | 44% | 65% |
> | **Fixed ρ=0.5** | 89% | 48% | 75% |
> | **Fixed ρ=0.7** | 82% | 38% | 65% |
> | **Smooth** | 88% | 46% | 80% |
> | **GAP (Ours)** | **91%** | **52%** | **85%** |
>
> The results in Table 2 demonstrate that employing LSTM for phase transition modeling significantly improves the quality of policy learning, **highlighting the necessity of this component within our estimation module**.
>
> Further, we conducted noise injection experiments by randomly selecting a direction and shifting each CPD output index with a probability of 0.5, repeating this shift process recursively. As shown in Table 3, even when considerable noise is injected into the CPD outputs, the policy performance remains robust. We attribute this robustness to the LSTM's capability to model continuous transitions, which **effectively alleviates the impact of potential errors introduced by the CPD algorithm**.
>
> **Table 3: Noise Injection Experiments**
> | Method | hammer | threading | put cube in drawer |
> |:------:|:------:|:---------:|:------------------:|
> | **Vision-only** | 86% | 44% | 70% |
> | **Concatenation** | 79% | 34% | 55% |
> | **Noise-Injected** | 88% | 49% | 85% |
> | **GAP (Ours)** | **91%** | **52%** | **85%** |
>
> We hope that the above ablation studies help to address your concerns, and we are happy to answer any further questions you may have.
>
> $\color{blue}Question \space 3: $ The extension of GAP to a fully automated system.
>
> $\color{darkgreen}Response \space 3: $
> We sincerely appreciate the reviewer's insightful question. First, we would like to clarify that our current GAP method is already fully automated and does not require manual labeling or human intervention. The motion transition phase estimation is performed automatically through our CPD algorithm and LSTM-based refinement, enabling the system to dynamically adjust gradient magnitudes during policy learning without any manual supervision.
>
> However, extending this approach to automatically determine when to pay more attention to proprioceptive signals versus visual cues presents significant challenges. As demonstrated by both the reported observations from Octo [4] and our results in Section 5.4, achieving effective implicit learning of attention mechanisms between visual and proprioceptive modalities through end-to-end training remains a complex task.
>
> One promising direction for future research would be utilizing the hidden states from our LSTM at each timestep as query, with features from both modalities serving as key and value. This could enable more explicit fusion mechanisms between the two modalities, potentially leading to dynamic and balanced collaboration.
>
> $\color{blue}Question \space 4: $ The lack of experiments with shared encoder.
>
> $\color{darkgreen}Response \space 4: $
> To the best of our knowledge, in the context of robotic manipulation, there has been limited exploration of using shared encoders for processing visual and proprioceptive information together. Given the focus of our research question on enhancing visual-proprioceptive collaboration, we chose to concentrate our experimental evaluation on the more commonly adopted dual-head architecture, which allows for clearer analysis of the interaction between the two modalities.
>
> However, as the reviewer correctly points out, in manipulation scenarios there exist various input modalities, such as depth images or vision-based tactile signals, which could potentially benefit from shared encoder architectures. We believe that investigating their collaboration in these scenarios would be a valuable contribution to the field.
>
> ---
>
> [1] Sun Q, Hong P, Pala T D, et al. Emma-x: An embodied multimodal action model with grounded chain of thought and look-ahead spatial reasoning. arXiv preprint arXiv:2412.11974, 2024.
>
> [2] Jia Z, Thumuluri V, Liu F, et al. Chain-of-thought predictive control. ICML, 2024.
>
> [3] Lu J, Xia W, Wang D, et al. Koi: Accelerating online imitation learning via hybrid key-state guidance. CoRL, 2024.
>
> [4] Team O M, Ghosh D, Walke H, et al. Octo: An open-source generalist robot policy. RSS, 2024.

---

> > ### Comment · Reviewer_RuJT · 2025-08-07
> >
> > I thank the authors for their efforts in providing the rebuttal. Most of my concerns have been addressed. However, the heuristic design of the CPD algorithm still remains a weak point of the paper, although empirically it's shown to provide better results. In particular, I'm curious about the definition of a motion transition phase. The authors mentioned that human annotation performs worse than the CPD because humans tend to favor semantically annotated results. But without a definition of a ground truth, how can one be sure this is the case? Secondly, without a clear definition of the motion transition phase, how can one design a principled approach (compared to heuristic CPD) to address the gradient imbalance issue?
> >
> > Nevertheless, this paper offers a new perspective on why robot policy performance may decline with the inclusion of proprioceptive signals. In addition, it proposes a solution to mitigate such an effect. As a result, I will maintain my rating.

---

> ### Author Response · Authors · 2025-08-04
>
> Dear reviewer, we would be grateful to receive your feedback regarding whether our responses have satisfactorily addressed the your concerns. Please let us know if there are any areas that would benefit from further discussion.

---

> ### Author Response · Authors · 2025-08-07
> **Thank you for your response**
>
> Thank you for your valuable suggestions and feedback. We also greatly appreciate the recognition of our efforts in addressing gradient imbalance issue in vision-proprioception policy learning for robotic manipulation.
>
>
>
> That's a good question regarding determining the ground truth for the motion transition phase. However, finding the best annotation for model learning is challenging, as the knowledge gained through learning often differs from human intuition. The experimental result showing that human annotations perform worse than the CPD algorithm further supports this idea. Instead of manually designing the ground truth transition phase, we decided to incorporate our prior knowledge into the automatic algorithm to generate suitable pseudo-labels for the models. For instance, our CPD algorithm is a dynamic programming method designed to find a global optimal solution for a target objective. Therefore, we can modify the prior objective to address the gradient imbalance issue.
>
>
>
> We hope our paper provides valuable insights for advancing robotic policy learning, particularly with the inclusion of proprioception.

---

### Official Review · Reviewer_KStE · 2025-07-02

**Clarity:** 3
**Significance:** 3
**Originality:** 2
**Rating:** 4
**Confidence:** 3

**Summary:**

The paper claims that adding proprioceptive information to VLA policies tends to hurt performance. The authors hypothesize this is because proprioceptive information is more useful when continuing the same skill, but visual information is more relevant when transitioning between skills, and the latter is down-weighted by learning algorithms in these transition phase due to the relatively subtler changes in feature values. They attempt to mitigate this by increasing the relative weighting of the gradients from visual features during motion-transition phases, which they define using a dynamic programming based change point detection algorithm on the end-effector state.
The approach is compared with previous ones on a series of manipulation benchmark tasks and show improved task success rate. Additional experiments also demonstrate an enhanced usage of the vision modality, and a compatibility with other VLA models and modality fusion approaches.

**Questions:**

1. How do you reconcile the fact that previous literature is mixed on whether proprioception hurts performance and your statement that it is a real problem? I am not familiar with this as a problem in the field so definitive evidence of this could change my evaluation.

2. Why do you train a network to predict transitions indicators if you are already labeling your training set with them.

**Ethical Concerns:**

["NO or VERY MINOR ethics concerns only"]

**Final Justification:**

Authors have clarified most of my concerns raised.

**Limitations:**

Yes

**Paper Formatting Concerns:**

No concerns.

**Quality:**

2

**Strengths And Weaknesses:**

Strengths: This paper presents a clear narrative, with a well-defined problem, a well reasoned mechanism causing this problem based on previous work, and a clear and simple solution to address this. They present a very comprehensive set of experiments to test the efficacy and applicability of their method with experiments in real and in sim. The questions answered in the experiments section are clear and well-motivated.
Weaknesses: The most significant weakness of this paper is the importance of the motivating problem and the marginality of the benefit it produces. In the authors’ own related work, they state that previous literature has mixed opinions of whether this is a real problem. The results in Table 1 show only minor increases in success for most tasks, possibly not being statistically significant enough to justify the importance of the underlying problem. In addition to this the paper has several grammatical errors, including the title, which could be changed to “when would visual-proprioceptive policies fail for robotic manipulation”. Other mistakes are “To support our analyze” on line 120 and the title for section 5.2.

---

> ### Author Rebuttal · Authors · 2025-07-31
>
> We sincerely appreciate the reviewer’s critical feedback and the time taken to evaluate our work. We take your concerns seriously and address each in detail below.
>
> $\color{blue}Question \space 1: $
> The existence of our core research problem. (clarifications and definitive evidences)
>
> $\color{darkgreen}Response \space 1: $
> We sincerely appreciate the reviewer's valuable feedback regarding the significance of our research problem. We acknowledge that the importance of visual-proprioceptive collaboration in robotic manipulation has indeed been subject to debate in the literature. However, our comprehensive experimental evaluation provides compelling evidence that this is indeed a meaningful and significant problem that warrants attention, which is also acknowledged and recognized by other reviewers.
>
> **Clarification of previous works:**
> Previous works have demonstrated that introducing proprioceptive information could harm policy performance. For instance, Octo [1] reported that policies trained with additional proprioception generally performed worse than vision-only policies. Similarly, RoboMimic [2] observed that the introduction of proprioceptive information led to a 2%-29% relative performance drop compared to vision-only policies. Meanwhile, HPT [3] enhances the robotic manipulation policy learning by incorporating proprioceptive information with a hard-coding design. However, there is a lack of comprehensive exploration on the impact of proprioceptive information on visual-proprioceptive policy learning. Thus, we performed an in-depth analysis and conducted extensive experiments to systematically investigate it as below.
>
> **Comprehensive problem validation:**
> To rigorously validate the existence of our identified problem, we designed large-scale experiments that systematically consider both the variety of policy architectures as well as the diversity and complexity of tasks. This comprehensive evaluation is crucial for establishing the generality and significance of the problem.
>
> - **Policy architecture variety:** Our evaluations span multiple policy architectures, including **MLP-based policies** and **diffusion-based policies**. This problem is further confirmed by both reported observations of Octo and our results in Section 5.4, demonstrating that this problem persists across **transformer-based policies** as well.
>
> - **Task complexity and diversity:** Our comprehensive evaluations encompass 14 diverse manipulation tasks, spanning articulated object manipulation (hammer), contact-rich interactions (push-wall, threading, handover), rotation-sensitive tasks (use rag to sweep table, lift lid and pour), and soft object manipulation (put thermos into bag). The problem manifests consistently across all these task categories.
>
> - **Consistent observations:** Across all experimental settings and configurations, vision-proprioception policies consistently underperform compared to vision-only policies. The application of our GAP method consistently leads to superior performance, providing strong evidence that this problem is indeed significant and widespread in visual-proprioceptive policy learning.
>
> **Statistical significance:**
> Each task in our experiments in Meta-World and RoboSuite is evaluated with 100 rollouts. These relatively large numbers of rollouts ensure the statistical significance of our results, **which is further strengthened by the multiple seeds results provided in Section B.1 of the supplementary materials**. After applying our proposed GAP method, visual-proprioceptive policies achieved an average performance improvement of 26.5%, with some tasks showing even more dramatic improvements (e.g., the "threading" task improved from 34% to 52%, providing a 53% relative improvement). This demonstrates the significant importance of addressing this problem for visual-proprioceptive policy learning.
>
> Our comprehensive evaluation provides strong evidence for the significance of this research problem, which is also acknowledged and recognized by other reviewers. We hope our response addresses your concerns, and we are happy to answer any further questions you may have.
>
> $\color{blue}Question \space 2: $ Roles of LSTM in Motion-Transition Phase Estimation module.
>
> $\color{darkgreen}Response \space 2: $
> We acknowledge that our brief introduction to the LSTM component may have caused some confusion, and we will provide a detailed explanation of its function along with experimental validation of its effectiveness.
>
> In our proposed GAP framework, we first define the robot's motion representation and motion-consistency distance, then utilize the Change Point Detection (CPD) algorithm to identify a set of indices that decompose the trajectory into motion-consistent phases. The output of CPD is a set of indices (for example, [I_1=10, I_2=25, I_3=53, ...]), meaning that for each timestep, we obtain a binary 0-1 label indicating whether the robot's motion is transitioning. However, transition is a continuous process, and **the discrete labels from CPD cannot capture this continuous nature effectively**. For example, if CPD outputs I_2=25, it cannot model the transition process at nearby timesteps such as 23, 24, or 26, which may also be part of the transition phase.
>
> Therefore, we further utilize a temporal network such as LSTM to better capture this continuous process. Through the LSTM, we estimate the probability that each timestep belongs to motion-transition phases. In other words, **the LSTM outputs continuous values in the range of 0-1**, which provides a more sophisticated modeling of the transition process and helps eliminate potential errors introduced by the CPD algorithm.
>
> To validate the effectiveness of the LSTM component, we conducted comprehensive ablation studies on the LSTM. Specifically, we compared the LSTM-based phase transition modeling with the following alternatives:
>
> - **Fixed ρ:** For the phase transition indices output by CPD, we applied a fixed gradient adjustment magnitude ρ to the corresponding samples during policy learning.
>
> - **Smooth:** Following KOI [4], we modeled the transition process by treating the CPD output index as the mean of a Gaussian distribution.
>
> **Table 1: Ablation Studies on the LSTM component**
> | Method | hammer | threading | put cube in drawer |
> |:------:|:------:|:---------:|:------------------:|
> | **Vision-only** | 86% | 44% | 70% |
> | **Fixed ρ=0.3** | 77% | 44% | 65% |
> | **Fixed ρ=0.5** | 89% | 48% | 75% |
> | **Fixed ρ=0.7** | 82% | 38% | 65% |
> | **Smooth** | 88% | 46% | 80% |
> | **GAP (Ours)** | **91%** | **52%** | **85%** |
>
> The results in Table 1 demonstrate that while alternative methods such as smoothing can serve as substitutes for modeling the transition process, they do not provide significant performance improvements.
>
> Further, to demonstrate the LSTM's effectiveness in eliminating potential estimation errors, we conducted noise injection experiments by randomly selecting a direction and shifting each CPD output index with a probability of 0.5, repeating this shift process recursively. As shown in Table 2, even when considerable noise is injected into the CPD outputs, the policy performance remains robust. We attribute this robustness to the LSTM's capability to model continuous transitions, which effectively alleviates the impact of errors introduced by the CPD algorithm.
>
> **Table 2: Noise Injection Experiments**
> | Method | hammer | threading | put cube in drawer |
> |:------:|:------:|:---------:|:------------------:|
> | **Vision-only** | 86% | 44% | 70% |
> | **Concatenation** | 79% | 34% | 55% |
> | **Noise-Injected** | 88% | 49% | 85% |
> | **GAP (Ours)** | **91%** | **52%** | **85%** |
>
> We thank the reviewer for this valuable suggestion. In the revised version, we will provide more detailed explanations and experiments to help readers better understand the design details of this module.
>
> $\color{blue}Question \space 3: $ Grammatical errors in the paper.
>
> $\color{darkgreen}Response \space 3: $
> We thank the reviewer for identifying grammatical errors and suggesting improvements to our presentation. We agree and will refine writings and correct the remaining errors in the revised version.
>
> ---
>
> [1] Team O M, Ghosh D, Walke H, et al. Octo: An open-source generalist robot policy. RSS, 2024.
>
> [2] Mandlekar A, Xu D, Wong J, et al. What Matters in Learning from Offline Human Demonstrations for Robot Manipulation. CoRL, 2022.
>
> [3] Wang L, Chen X, Zhao J, et al. Scaling proprioceptive-visual learning with heterogeneous pre-trained transformers. NeurIPS, 2024.
>
> [4] Lu J, Xia W, Wang D, et al. Koi: Accelerating online imitation learning via hybrid key-state guidance. CoRL, 2024.

---

> ### Author Response · Authors · 2025-08-04
>
> Dear reviewer, we would appreciate knowing if our responses have fully addressed your concerns. We are happy to answer any further questions or comments you may have.

---

> > ### Comment · Reviewer_KStE · 2025-08-06
> >
> > Thank you for the rebuttal and for addressing most of my feedback with constructive and detailed responses. Most of my questions have been answered or clarified.

---

> > > ### Author Response · Authors · 2025-08-06
> > > **Thank you for your acknowledgement**
> > >
> > > Thank you for the helpful suggestions and recognition. We will incorporate the clarifications and additional results into the revised version of our paper.

---

### Official Review · Reviewer_czo3 · 2025-07-03

**Clarity:** 3
**Significance:** 3
**Originality:** 3
**Rating:** 4
**Confidence:** 4

**Summary:**

In this work, we attempt to answer the question: When would vision-proprioception policy fail? The authors conduct temporally controlled experiments and found that during task sub-phases that robot’s motion transitions, which require target localization, the vision modality of the vision-proprioception policy fails to take effect and concise proprioceptive signals that offer faster loss reduction when training, thereby dominating the optimization and suppressing the learning of the visual modality during motion-transition phases. The paper proposes the Gradient Adjustment with Phase-guidance (GAP) algorithm.  The authors conduct  comprehensive experiments demonstrate GAP is applicable in both simulated and real-world environments, across one-arm and dual-arm setups, and compatible with both conventional and Vision-Language-Action models.

**Questions:**

1. What is the relationship between the proposed method with guided diffusion policy [1]? While they are not the same, it seems to me there are some similarity between the two ideas. It would be great to have a connect them. Can guided diffusion be another implementation to realize the same concept as in this work?
[1] Xu et al., Dynamics-Guided Diffusion Model for Robot Manipulator Design, CoRL 2024

2. What is the standalone performance of this estimation module?

3. How sensitive is the system to estimation errors?

4. If ground truth phase labels were provided, how would the policy performance differ?

**Ethical Concerns:**

["NO or VERY MINOR ethics concerns only"]

**Final Justification:**

I appreciate the authors’ efforts in addressing the concerns raised in my original review. Most of the issues have been adequately addressed. However, I remain unconvinced by the response to the first question regarding the overgeneralization of the core research question. Additionally, the current title of the paper appears to be beyond the actual scope of the work. I recommend narrowing the scope and revising the title to more accurately reflect the specific contributions and experiments presented in the paper.

**Limitations:**

Yes

**Quality:**

3

**Strengths And Weaknesses:**

**Strengths**

1. The question of when to rely on vision versus proprioception signals during policy training is a critical and timely challenge in the robot learning community. The visualizations in Figures 1 and 4 effectively highlight this problem and provide compelling motivation.
2. To address this, the authors introduce a novel algorithm Gradient Adjustment with Phase-guidance (GAP), which adaptively modulates proprioceptive optimization. This approach enables dynamic and context-aware collaboration between visual and proprioceptive inputs within a unified policy.
3. Notably, GAP is compatible with both conventional policy architectures and emerging Vision-Language-Action models. Its versatility and effectiveness are well-supported by extensive experiments across both simulated and real-world environments.

**Weaknesses**

Overall, this work presents a promising direction for advancing multimodal policy learning in robotic manipulation. However, several critical points require clarification and refinement before the contributions can be fully appreciated and appropriately scoped.

1. Overgeneralization of the Core Research Question: The central question posed—“When do vision-proprioception policies fail in robotic manipulation?”—is framed too broadly given the scope of the study. The vision-proprioception policies evaluated in this work represent only a subset of the broader policy design space, excluding other relevant architectures such as diffusion-based policies. Furthermore, the experimental tasks are limited in complexity and diversity; contact-rich interactions and articulated object manipulation, both of which pose unique challenges to vision and proprioception integration, are not addressed. As such, the authors are strongly encouraged to revise the framing of their research question and claims to better reflect the actual coverage and limitations of the current study.
2. Lack of Analysis on Motion-Transition Phase Estimation Robustness: The Motion-Transition Phase Estimation module appears to be a critical component of the proposed GAP framework, as it governs when and how proprioception is emphasized during training. However, the paper lacks an in-depth analysis of its accuracy and impact on overall performance. Specifically:
    - What is the standalone performance of this estimation module?
    - How sensitive is the system to estimation errors?
    - If ground truth phase labels were provided, how would the policy performance differ?

Including such an ablation or comparison would greatly enhance the reader’s understanding of the module’s influence on the final results and provide insights into the robustness and generalizability of the proposed method.

---

> ### Author Rebuttal · Authors · 2025-07-31
>
> Thanks for dedicating time and effort to review our work and providing valuable feedback. We have thoroughly considered all your comments and respond to each below:
>
> $\color{blue}Question \space 1: $
> The overgeneralization of the core research question.
>
> $\color{darkgreen}Response \space 1: $
> We appreciate the reviewer's valuable feedback regarding the scope of our research question. We acknowledge that comprehensive evaluation across diverse policy architectures and task complexities is essential to thoroughly address our central research question. We would like to clarify the generalization of our core research question below.
>
> - **Policy Architecture Coverage:** We have evaluated **MLP-based policies** in the main paper along with **diffusion-based policies in Section B.2 of the supplementary materials**. Our experimental results demonstrate that while diffusion-based policies generally outperform MLP-based ones, concatenation-based vision-proprioception policies still underperform compared to vision-only policies. Furthermore, both reported observations of Octo [1] and our results in Section 5.4 confirm that this issue persists across **transformer-based policies** as well.
>
> - **Task Complexity and Diversity:** As the reviewer points out, experiments across tasks of various complexity and diversity are crucial for demonstrating the generalization of our question. Our evaluations comprehensively cover a wide range of manipulation tasks, including articulated object manipulation (hammer), contact-rich interactions (push-wall, threading, handover), rotation-sensitive tasks (use rag to sweep table, lift lid and pour), and soft object manipulation (put thermos into bag). Detailed descriptions of all tasks are provided in Section C of the supplementary materials.
>
> - **Consistent Observations:** Across the evaluated manipulation scenarios, vision-proprioception policies consistently underperform compared to vision-only policies. After equipped with GAP, policies demonstrate superior performance, highlighting the significance of addressing this fundamental challenge in vision-proprioception policies.
>
> We thank the reviewer for this constructive feedback and will provide more detailed explanations in the revised version to enhance reader understanding of our core research question.
>
> $\color{blue}Question \space 2: $ The absence of analysis of Motion-Transition Phase Estimation module. (the standalone performance and comparisons with human annotations)
>
> $\color{darkgreen}Response \space 2: $
> In our proposed GAP framework, the Motion-Transition Phase Estimation module is essential for addressing the core research question. We value your feedback and recognize that an in-depth analysis of this module would be important in enhancing the quality of our paper. To thoroughly analyze its effectiveness and robustness, we additionally conducted comprehensive ablation studies.
>
> First, we kept all other components unchanged and tested the impact of different trajectory decomposition methods on policy learning. The detailed descriptions of the comparison methods are as follows:
>
> - **Human:** Since obtaining ground truth phase transition labels is infeasible due to the complexity and diversity of manipulation tasks, we instead employ human experts to manually label phase transitions.
>
> - **HDBSCAN:** Following Emma-X [2], we utilize the HDBSCAN algorithm to cluster our distance metric defined in Equation 4, decomposing trajectories into phases.
>
> - **CoTPC [3]:** To decompose task trajectories into temporally close and functionally similar subskills, CoTPC applies the Change Point Detection (CPD) algorithm on the sequence of robot actions using Cosine distance.
>
> **Table 1: Performance Comparison between GAP and other decomposition methods**
> | Method | hammer | threading | put cube in drawer |
> |:------:|:------:|:---------:|:------------------:|
> | **Vision-only** | 86% | 44% | 70% |
> | **Human** | 88% | 47% | 75% |
> | **HDBSCAN** | 82% | 37% | 65% |
> | **CoTPC** | 85% | 50% | 85% |
> | **GAP (Ours)** | **91%** | **52%** | **85%** |
>
> As shown in Table 1, using human-labeled phase transitions for gradient adjustment does provide a slight improvement in performance. However, since human annotators tend to decompose trajectories from a semantic perspective rather than based on actual motion changes, this approach may miss certain transition points, resulting in only limited gains. The cluster-based HDBSCAN method disrupts the temporal structure inherent to manipulation tasks, leading to even worse policy performance compared to human labeling. Notably, both CoTPC and our GAP framework employ the CPD algorithm for phase estimation and generally outperform human annotation across most tasks, suggesting that automated phase estimation can capture motion transitions more effectively than manual labeling. However, CoTPC relies on a simple cosine distance, which lacks the ability to represent the nuanced characteristics of robot motion, thus yielding less significant performance improvements compared to GAP. In our approach, the motion-consistent distance defined in Equation 4 captures spatial position changes, rotational changes, and gripper opening state changes, enabling more accurate and efficient prediction of motion transitions.
>
> To further understand **the performance gap between GAP and human labeling**, we calculated the precision and recall between GAP outputs and human labels. Our results show that GAP achieves a high recall (97% on average) but a relatively lower precision (73% on average), suggesting that the CPD algorithm not only accurately detects semantic changes during manipulation but also identifies motion transitions that may be overlooked by humans. **This experiment highlights that our method offers an effective and efficient approach, given that obtaining ground truth phase transition labels is infeasible due to the complexity and diversity of manipulation tasks.**
>
> We thank the reviewer for this essential feedback. In the revised version, we will incorporate the above ablation studies and detailed analysis of the Motion-Transition Phase Estimation module to better demonstrate its effectiveness and importance within our framework.
>
> $\color{blue}Question \space 3: $
> How sensitive is the system to estimation errors?
>
> $\color{darkgreen}Response \space 3: $
> We appreciate the reviewer's concern regarding the system's sensitivity to estimation errors. To thoroughly address this important question, we conducted systematic experiments to assess whether estimation errors from the CPD algorithm would significantly impact policy learning. Specifically, we injected noise into the CPD outputs by randomly selecting a direction and shifting each index with a probability of 0.5, repeating this shift process recursively. As shown in Table 2, even when considerable noise is injected, the policy performance remains robust, indicating that our system exhibits strong tolerance to estimation errors. We attribute this robustness to the LSTM's capability to model continuous transitions, which effectively alleviates the impact of errors introduced by the CPD algorithm.
>
> **Table 2: Noise Injection Experiments**
> | Method | hammer | threading | put cube in drawer |
> |:------:|:------:|:---------:|:------------------:|
> | **Vision-only** | 86% | 44% | 70% |
> | **Concatenation** | 79% | 34% | 55% |
> | **Noise-Injected** | 88% | 49% | 85% |
> | **GAP (Ours)** | **91%** | **52%** | **85%** |
>
> $\color{blue}Question \space 4: $
> The relationship between GAP with guided diffusion policy [5].
>
> $\color{darkgreen}Response \space 4: $
> We thank the reviewer for raising this interesting question about the relationship between our GAP framework and guided diffusion policy work. We would like to clarify the key differences and potential connections between these two ideas.
>
> The guided diffusion policy work focuses on embodiment intelligence, aiming to generate manipulator designs for given manipulation tasks. They utilize the loss between the predicted interaction profile and the target profile, and the backpropagated gradients to guide the diffusion model in generating manipulators with suitable shapes. Therefore, the diffusion model in that work is not involved in policy learning.
>
> However, we acknowledge that given the powerful distribution modeling capability of diffusion models, it represents an interesting and potentially promising direction to use information such as the sequence of proprioception in task trajectories as conditions for diffusion models to estimate motion transitions phase, which could benefit the learning of vision-proprioception policies. This could be a valuable avenue for future research that builds upon our current work.
>
> ---
>
> [1] Team O M, Ghosh D, Walke H, et al. Octo: An open-source generalist robot policy. RSS, 2024.
>
> [2] Sun Q, Hong P, Pala T D, et al. Emma-x: An embodied multimodal action model with grounded chain of thought and look-ahead spatial reasoning. arXiv preprint arXiv:2412.11974, 2024.
>
> [3] Jia Z, Thumuluri V, Liu F, et al. Chain-of-thought predictive control. ICML, 2024.
>
> [4] Lu J, Xia W, Wang D, et al. Koi: Accelerating online imitation learning via hybrid key-state guidance. CoRL, 2024.
>
> [5] Xu et al. Dynamics-Guided Diffusion Model for Robot Manipulator Design. CoRL, 2024.

---

> ### Author Response · Authors · 2025-08-04
>
> Dear reviewer, we would like to know whether our responses addressed your concern. We are happy to answer any further questions and comments.

---

> ### Comment · Reviewer_czo3 · 2025-08-07
>
> I appreciate the authors’ efforts in addressing the concerns raised in my original review. Most of the issues have been adequately addressed. However, I remain unconvinced by the response to the first question regarding the overgeneralization of the core research question. Additionally, the current title of the paper appears to be beyond the actual scope of the work. I recommend narrowing the scope and revising the title to more accurately reflect the specific contributions and experiments presented in the paper.
>
> A side note: I would not classify “hammer” as an example of articulated object manipulation. In contrast, tasks such as “opening the door of a cabinet” fall more clearly under this category. Please clarify this distinction in the manuscript.

---

> ### Author Response · Authors · 2025-08-07
> **Thank you for your response**
>
> We sincerely thank the reviewer for the thoughtful comments and constructive suggestions.
>
> Regarding the concern about the overgeneralization of our core research question, we have conducted extensive experiments across a wide range of manipulation tasks to support the generality of the observed phenomenon. Nonetheless, we acknowledge the importance of clearly defining the scope of our contributions and will consider to revise the title and presentation of the paper to better reflect the specific contributions and experiments presented in our paper.
>
> As for the task "hammer", we agree that this term may have caused confusion. In this task, the robot first opens a drawer, which involves articulated object manipulation, then proceeds to retrieve and place the hammer into drawer, and finally closes the drawer. We apologize for the unclear description and will revise it in the final version to ensure that readers correctly understand this task.

---

### Note · Authors · 2025-08-14

We sincerely thank the reviewers and the Area Chair for their valuable feedback and constructive discussions, which have greatly helped refine the clarity and quality of our work.

In this study, we demonstrate that vision-proprioception policies can fail during motion-transition phases due to the suppression of the vision modality. To address this challenge, we propose the Gradient Adjustment with Phase-guidance (GAP) algorithm, which enables dynamic collaboration between vision and proprioception within such policies. We believe our work provide meaningful insights for advancing vision-proprioception policies in robotic manipulation.

We greatly appreciate the reviewers' recognition of our work, including the importance of the research question, the hypothesis-driven approach, the extensive experiments conducted in both simulation and the real world, and the broad applicability of our algorithm. We are also thankful for the reviewers' constructive suggestions, such as refining the presentation of the research question, performing additional ablation studies on the GAP, and providing a more detailed description of the experimental settings.

We are encouraged that our rebuttal has addressed the reviewers' concerns. The additional experiments and improved presentation will be incorporated into the revised version to further strengthen the paper. Again, we sincerely thank the reviewers and the Area Chair for their time, insightful comments, and constructive guidance.

---

### Decision · Program_Chairs · 2025-09-17

**Decision:**

Reject

**Comment:**

The paper addresses visuo-proprioceptive fusion in robotic manipulation. Its key observation is that the importance of each modality varies across task sub-phases. To improve the fusion, the paper proposes the Gradient Adjustment with Phase-guidance algorithm (GAP) that adaptively modulates the gradient of proprioception modality during training.

The main concerns from the reviewers include:
1. Overgeneralization of the core research question (czo3);
2. Limited analysis of the phase estimation, e.g., the standalone performance of the estimation module, sensitivity to estimation errors, and the upper bound of using ground-truth labels (czo3, KStE, RuJT);
3. Limited discussion on previous literature with mixed opinions regarding whether proprioception hurts performance (KStE);
4. Unclear definition of motion transition and heuristic design of the CPD algorithm (RuJT);
5. Insufficient details in key experimental setups (48PM);
6. OOD evaluation limited to object placements; lacking broader visual distribution shifts (48PM);
7. Concerns that ignorance of vision inputs might not be an issue for models with strong vision encoders (48PM).

The AC has carefully checked the reviewers' comments and the rebuttal. The rebuttal addressed most of these concerns; however, there are remaining concerns:
- Scope (point 1): The AC agrees with Reviewer czo3 that the research question is overstated, as the experiments lack task diversity especially for dexterous manipulation [1]. In dexterous manipulation, the use of proprioception is standard; such tasks should be included instead of focusing only on gripper-based manipulation.
- Literature discussion (point 3): The authors should discuss more work using proprioception in robotic manipulation [1,2]. Many existing work simply uses concatenation or addition to fuse visual-proprioceptive information, but all the tasks evaluated in the paper do not benefit from this simple multimodal fusion approach. More discussions are needed.
- Motion phases design (point 4): Reviewer RuJT remains less convinced by the heuristic design of motion phases.
- Points 6 and 7: Generalization and using strong vision backbones are important. The rebuttal provided some results on selected tasks. However, the AC finds the evaluation insufficient, as the results and ablations are a standard setting where no generalization is required. It remains unclear whether conclusions hold with stronger vision encoders or more diverse tasks.
- Multimodal fusion baselines: the AC also considers that more multimodal fusion methods should be compared (e.g., attention, gating mechanisms) against GAP.

Although the reviewers’ ratings were 2 Borderline Accepts and 2 Accepts, the AC finds substantial room for revision. Given the high bar of NeurIPS, the AC recommends rejection. The authors are encouraged to strengthen the scope, methodology, and evaluation and resubmit to a next venue.

[1] Qin Y, Huang B, Yin Z H, et al. Dexpoint: Generalizable point cloud reinforcement learning for sim-to-real dexterous manipulation[C]//Conference on Robot Learning. PMLR, 2023: 594-605.

[2] Ze Y, Zhang G, Zhang K, et al. 3d diffusion policy: Generalizable visuomotor policy learning via simple 3d representations[J]. arXiv preprint arXiv:2403.03954, 2024.